# Exploring the Biological Activity and Mechanism of Xenoestrogens and Phytoestrogens in Cancers: Emerging Methods and Concepts

**DOI:** 10.3390/ijms22168798

**Published:** 2021-08-16

**Authors:** Xiaoqiang Wang, Desiree Ha, Ryohei Yoshitake, Yin S. Chan, David Sadava, Shiuan Chen

**Affiliations:** Department of Cancer Biology, Beckman Research Institute, City of Hope, 1500 E. Duarte Rd., Duarte, CA 91010, USA; xiaoqiwang@coh.org (X.W.); deha@coh.org (D.H.); ryoshitake@coh.org (R.Y.); ychan@coh.org (Y.S.C.); dsadava@coh.org (D.S.)

**Keywords:** cancer, endogenous estrogens, estrogen receptors, exogenous estrogens, patient-derived xenograft/PDX, phytoestrogens, single-cell RNA sequencing/scRNA-seq, window of susceptibility/WOS, xenoestrogens

## Abstract

Xenoestrogens and phytoestrogens are referred to as “foreign estrogens” that are produced outside of the human body and have been shown to exert estrogen-like activity. Xenoestrogens are synthetic industrial chemicals, whereas phytoestrogens are chemicals present in the plant. Considering that these environmental estrogen mimics potentially promote hormone-related cancers, an understanding of how they interact with estrogenic pathways in human cells is crucial to resolve their possible impacts in cancer. Here, we conducted an extensive literature evaluation on the origins of these chemicals, emerging research techniques, updated molecular mechanisms, and ongoing clinical studies of estrogen mimics in human cancers. In this review, we describe new applications of patient-derived xenograft (PDX) models and single-cell RNA sequencing (scRNA-seq) techniques in shaping the current knowledge. At the molecular and cellular levels, we provide comprehensive and up-to-date insights into the mechanism of xenoestrogens and phytoestrogens in modulating the hallmarks of cancer. At the systemic level, we bring the emerging concept of window of susceptibility (WOS) into focus. WOS is the critical timing during the female lifespan that includes the prenatal, pubertal, pregnancy, and menopausal transition periods, during which the mammary glands are more sensitive to environmental exposures. Lastly, we reviewed 18 clinical trials on the application of phytoestrogens in the prevention or treatment of different cancers, conducted from 2002 to the present, and provide evidence-based perspectives on the clinical applications of phytoestrogens in cancers. Further research with carefully thought-through concepts and advanced methods on environmental estrogens will help to improve understanding for the identification of environmental influences, as well as provide novel mechanisms to guide the development of prevention and therapeutic approaches for human cancers.

## 1. Introduction

Estrogens are classified as either endogenous or exogenous, according to their origins [1]. Yet, both can bind to estrogen receptors (ERs), and/or many other nuclear receptors, simultaneously triggering genomic and transcriptomic changes in various organ systems. These changes can consequently contribute to the initiation and progression of multiple types of cancers, including the classical hormone-related breast and prostate cancer [2,3], as well as the non-classical hormone-related cancers, such as lung cancer [4], colorectal cancer [5], and gastric cancer [6]. 

Endogenous estrogens (estradiol/E2, estrone/E1, and estriol/E3) in humans are produced by endocrine glands and/or by extra-glandular tissues through steroidogenesis enzymes, such as cytochrome P450 oxidases (CYPs), hydroxysteroid dehydrogenases (HSDs), and aromatase (CYP19) [7]. Although the sex gonads (ovaries and testes) and adrenal cortex are the primary sites of estrogen synthesis, extra-gonadal estrogens are also produced in the mammary glands, lungs, liver, and intestine, and play an equally important role in controlling biological activities [8]. The important roles of endogenous estrogens in the etiology of breast cancer have been extensively studied, leading to the development of well-tolerated endocrine therapy for breast cancer [9]. 

Exogenous estrogens are those which are produced outside of the human body. In addition to synthetic estrogens developed for pharmacological purposes, a group of chemicals have been found to have estrogen-like activities, such as the ability to bind to ERs and to modulate the expression of estrogen-regulated genes. These exogenous and unexpected estrogen mimics include synthetic industrial compounds (xenoestrogens) and phytochemicals (phytoestrogens) [10]. They can alter the activities of ERs and send false signals, disrupting the normal estrogen response, changing physiological functions, and promoting diseases, including cancer [11]. Xenoestrogens include synthetic industrial chemicals used as solvents/lubricants and their byproducts such as plastics (bisphenol A, BPA), plasticizers (phthalates), flame retardants (polybrominated diphenyl ethers, PBDEs), pesticides (dichlorodiphenyltrichloroethane, DDT), and pharmaceutical agents (diethylstilbestrol, DES). The scientific consensus on xenoestrogens characterizes them as serious environmental hazards that have hormone-disruptive effects on both wildlife and humans [12]. Phytoestrogens are plant-produced compounds found in a wide variety of herbs and foods, most notably, soy-containing foods. Phytoestrogens, made naturally, often share structural features with endogenous E2, allowing phytoestrogens to cause estrogenic and/or anti-estrogenic effects [13]. They have been suggested to have a large spectrum of beneficial effects, including the reduction of cancer risk and postmenopausal symptoms [14]. However, there is also concern that phytoestrogens may act as endocrine disruptors that adversely affect health [15]. Based on available research findings, it is not clear whether the potential health benefits of phytoestrogens outweigh their risks. The potential for endocrine disruption by phytoestrogens needs to be considered as well [13]. Compared with endogenous estrogens, exogenous estrogens represent an under-recognized contributor to the development and progression of cancers. Further research on exogenous estrogens will help to provide insights for the identification of environmental influences, as well as provide new perspectives in the development of prevention and therapeutic approaches against human cancers.

At the molecular and cellular levels, xenoestrogens/phytoestrogens can imitate endogenous estrogens by enhancing and/or interrupting endogenous estrogen signaling pathways. They may exert either beneficial or harmful activities in humans depending on a set of complex factors such as exposure dose, time, intracellular signal transduction, and tissue complexity [16]. The binding of estrogens to ERs results in the activation of estrogen signaling pathways. There are intracellular ERs, including ER-alpha (ERα) and ER-beta (ERβ), as well as membrane-associated ERs, such as membrane ERs (mERs) and G Protein-Coupled Estrogen Receptors (GPER/GPR30) [17]. In addition to binding to ERs, exogenous estrogens can exert estrogenic activity by cross-talk with many other pathways, including pathways related to membrane-associated growth factor receptors, such as human epidermal growth factor receptor (EGFR/HER) and insulin-like growth factor 1-receptor (IGF1R) [18], as well as nuclear receptors, including aryl hydrocarbon receptor (AhR) [19], peroxisome proliferator-activated receptors (PPARs) [20], and estrogen-related receptor alpha/gamma (ERRα/γ) [21]. Multiple synergistic signaling pathways may contribute to the outcome of exogenous estrogen exposure on overall health and/or cancer cells. At the tissue level, exogenous estrogens may exhibit another dimension of complexity by influencing both cancer cells and cancer-associated stromal cells, including immune cells, fibroblasts, and adipocytes [22]. At the systemic level, exposure to exogenous estrogens has been linked to increased breast cancer risk during certain life stages known as the windows of susceptibility (WOS) including the prenatal, pubertal, pregnancy, and menopausal transition periods, during which the mammary glands undergo anatomical and functional transformations. Therefore, environmental hormones (e.g., endocrine-disrupting chemicals/EDC) and certain therapeutics (e.g., prescribed for the coexisting medical conditions or in the form of the hormone replacement therapy) can influence breast cancer risk, development, or outcome [23]. Considering the spatial heterogeneity (variety of cell types) and temporal heterogeneity (various stages of differentiation) of cancer, xenoestrogens/phytoestrogens could display integrated activities in a tumor-selective and/or life stage(s)-specific manner. 

The growing concerns of the exogenous estrogenic influence on health, especially towards cancer, have prompted considerable public attention and scientific interest. Knowledge of how these exogenous estrogens mimic endogenous estrogens, and how they exert their impacts on overall health, is crucial to resolve their impacts in the etiology of varying cancers. In this review, we conducted an exhaustive evaluation on the advanced research technology, molecular mechanisms, and ongoing translational studies in the development of prevention and therapeutic approaches towards human cancers. Here, we aim to provide thorough, updated understandings of xenoestrogens/phytoestrogens and their biological activities and mechanisms in cancer. 

## 2. Xenoestrogens and Phytoestrogens: Definitions and Origins 

### 2.1. Xenoestrogens: Synthetic Industrial Chemicals

Xenoestrogens are synthetic industrial chemicals found in various plastics, sealants, consumer goods, preservatives, and pesticides. They have unexpected activities by acting as either estrogen, triggering receptor pathways, or anti-estrogens, blocking normal estrogenic activity. These synthetic industrial chemicals can affect health and possibly trigger cancer [24]. The impact of these estrogen mimics is dictated by their binding affinities towards different types of ERs, predominantly ERα and ERβ, with ERα binding playing a pro-oncogenic role and ERβ typically playing a tumor-suppressive role [25] (Table 1).

An extensively studied xenoestrogen is bisphenol A (BPA). BPA was first used as a pharmaceutical estrogen in the 1930s but is now commonly used in the manufacture of polycarbonate plastics and epoxy resins used in food containers, water bottles, and other protective coatings [26]. BPA has been shown to disrupt ER activity by mimicking, enhancing, or inhibiting endogenous estrogens, causing a direct impact on the intracellular signal transduction pathways [27]. It has a relative binding affinity of 0.01 for both ERα and ERβ and has been strongly correlated with an increased risk for breast, prostate, and uterine cancer [28]. Because of this, many organizations concerned with the environment have suggested that the public avoid using items made with BPA [29]. 

Another xenoestrogen of interest is the estrogenic pesticide DDT which has been banned in the US for almost 50 years. DDT was a commonly used pesticide sprayed across many agricultural fields and homes, acting as an insect neurotoxin to kill mosquitoes and other insect vectors that carry malaria, typhus, and other insect-borne diseases. It is still widely used, particularly in India and southern Africa [30]. Only later would it be known that DDT accumulates in adipose tissue and continues to persist in the environment [31]. Adverse environmental effects on non-insect species led to DDT being banned in many countries. Since then, scientists have continued to study its estrogenic activity and its impacts on gene expression and hormone synthesis through transgenerational studies. With a relative binding affinity of 0–0.01 and 0–0.02 to ERα and ERβ, respectively, DDT was previously not associated with increased cancer risk. However, DDT has been linked to increased breast cancer, especially if the tissue is exposed during certain WOS [32]. Following the banning of DDT, methoxychlor (DMDT) was synthesized as an alternative for vector control. It was used to protect pets, crops, and livestock from pests such as mosquitoes, cockroaches, and other insects. Despite growing evidence that DMDT is an ERα agonist and ERβ antagonist, with relative binding affinities of <0.01% for both, resulting in increased inhibition of estrogen binding, it is still currently being used today. DMDT has been associated with increased ovarian cancer risk, but not with other human cancers [33].

Although numerous studies indicating the toxic effects of both BPA and DDT, these two xenoestrogens are still being used today. BPA has continued to be used in plastics despite epidemiological studies correlating its exposure with decreased sperm quality in males [34]. On the other hand, while DDT has been banned in the US for a half-century, it is still used in regions where malaria is endemic, is concerning as both epidemiological and clinical data have reported a decrease in semen volume, concentration, motility, and normal morphology to be associated with DDT exposure [35].

Polychlorinated biphenyls (PCBs) are well-known xenoestrogens that are widely used to make various electrical equipment, such as transformers and capacitors, and are also found in hydraulic fluids and plasticizers. These materials eventually make their way into landfills, where PCBs can re-enter the environment by being released into the soil and air [36]. PCBs include many compounds that have relative binding affinities for ERα and ERβ between 0.01 and 3.4 and <0.01 and 7.2, respectively. These relative binding affinities were adapted from Kuiper et al. who used solid-phase competition experiments to calculate binding affinities by setting E2 as 100 [37]. Despite the almost two-fold difference in ERβ binding affinity, compared to ERα, PCBs are associated with an increased breast cancer risk, making them a significant topic of research [38]. 

PBDEs are used as flame retardants, electrical equipment coating, construction materials, textiles, and furniture padding [39,40]. PBDEs encompass a large umbrella of compounds that have a relative binding affinity range of 1.3–20 to ERs [41]. Despite PBDEs exhibiting a higher binding affinity to ERs than that of PCBs, there has been no clear conclusion between PBDE exposure and breast cancer risk. Studies from our group recently demonstrated in breast cancer PDX models that PBDEs induced the expression of estrogen-responsive genes, especially genes related to cell proliferation in cancer cells [42,43]. 

Unlike the previously mentioned xenoestrogens, DES was synthesized as an “estrogen” and was previously prescribed to women to prevent miscarriages, premature labor, and pregnancy complications, before it, too, was realized to be carcinogenic. However, DES is no longer used to treat pregnancies at risk for miscarriage and menopausal symptoms but is still rarely used to treat prostate and breast cancer [44]. DES was the first synthetic estrogen and the first carcinogen to be shown to cross the placenta to cause cancer in the offspring. It has a potent relative binding affinity of 236% and 221% to ERα and ERβ, respectively, due to the additional hydrophobic interactions causing DES to be a potent transcriptional activator through genomic signaling [45]. 

Ethinyl estradiol (EE2) is a xenoestrogen synthetically derived from E2. It works as an ovulation inhibitor and is mostly found in hormonal contraceptives. It has a strong relative binding affinity of 190% for ER and has been shown to increase cell proliferation, but at a lower rate than E2. There have been controversial data regarding EE2’s effects on cancer risk, but more recent studies have suggested that EE2 has little/no breast cancer risk, while having decreased ovarian, endometrial, colorectal, lymphatic cancer risks [46].

Other xenoestrogens of interest include phthalates, nonylphenols (NP), and parabens. Phthalates are found in soft packaging plastic materials and can competitively inhibit E2 binding to ER [47,48]. Meanwhile, NP is used in various industrial processes and is found in consumer goods, such as laundry detergents, personal hygiene, automotive, and lawn care products. NP has a low relative binding affinity to ER of 0.0032–0.037, compared to the relative binding affinities of other xenoestrogens. Even so, they can exhibit an estrogen-like activity on ER^+^ breast cancer cells [49,50]. On the other hand, parabens are preservatives used in many consumable items such as beer, sauce, soda, and several cosmetics. They have a relative binding affinity range of 0.011–0.11 and 0.011–0.123 for ERα and ERβ, respectively, and can increase breast cancer cell proliferation and tumor size in animals [51,52]. These three types of xenoestrogens have all been implicated with breast cancer risk and their continued presence jeopardizes future health standards.

The presence of xenoestrogens in our environment and our everyday products warrants more research into their implications concerning cancer. Although some compounds have lower binding affinities than others, their impact on ERα, as well as their increased cancer risks, necessitates more attention to understanding the exact mechanisms and route of exposure by which they function.

### 2.2. Phytoestrogens: Plant-Derived Chemicals

Phytoestrogens are a group of estrogen mimics present in plants. They are becoming subjects of interest due to their estrogenic potentials and constant exposure to humans (Table 2).

Soybeans, a staple in many Asian cuisines, contain two major isoflavones: genistein (GEN) and daidzein (DAI). Although similar in structure and function, GEN has both stronger binding to ERβ than ERα. For GEN, the difference is 20-fold, and for DAI the difference is five-fold. This stronger binding affinity for ERβ, combined with the observation that GEN results in a decrease of ERα mRNA and protein levels [53,54,55,56,57,58], has led to clinical trials in cancer prevention and treatment. To date, GEN and DAI have been shown to reduce breast cancer-related gene expression [59,60], and reduce the increase in serum PSA during prostate cancer development [61]. Both GEN and DAI are well tolerated with minimal toxicity. 

Other phytoestrogens, such as quercetin (QUE) [62,63,64,65,66,67,68], apigenin (APE) [69,70,71,72,73,74], resveratrol (RES) [75,76,77,78,79], myricetin (MYR) [80,81,82,83,84,85], and are found in many berries, leafy greens, and wine. Although their relative binding affinity differences between ERα and ERβ are not as great as in GEN and DAI, these compounds have been investigated due to their widespread presence in plants and extensive human consumption. More specifically, QUE, APE, and even RES have been noted to exhibit a biphasic effect; at low concentrations, these phytoestrogens display estrogenic activity, whereas, at higher concentrations, they display more protective anti-estrogenic activity [62,63,64,65,66,67,68,69,70,71,72,73]. Like GEN, RES has been extensively studied in many clinical trials. It has been shown that RES can significantly decrease epigenetic gene methylation in women at high risk for breast cancer and suppresses the important WNT signaling pathway [75,76,77,78,79]. These findings support the chemo-preventive effects of RES as possible cancer therapeutic. However, health beneficial effects of RES have not been established due to non-physiological research designs.

Kaempferol (KPF), found in tea, pollen, and garlic, has been shown to decrease breast cancer risk possibly due to its 30-fold difference in ERα and ERβ relative binding affinities. KPF, although fairly novel, is an exciting phytoestrogen due to its ability to decrease cancer cell growth and increase apoptosis [86,87,88,89,90,91]. On the other hand, luteolin (LUT) is another more recently studied phytoestrogen found in seasonings that exhibits similar results as KPF: increasing cell cycle arrest, apoptosis, and decreasing proliferation [92]. 

Of all the reviewed phytoestrogens, curcumin (CUR) has been the most evaluated in terms of both pre-clinical and clinical investigations. CUR is derived from the plant *Curcuma longa*, otherwise known as turmeric. In breast cancer cells and tissues exposed to CUR, it has been shown to decrease ER expression, leading to decreased cell proliferation, migration, invasion, and angiogenesis, while increasing apoptosis, cell cycle arrest, and senescence in breast cancer cell lines [93,94,95]. In clinical trials, CUR has been shown to slightly reduce fatigue in women with advanced, metastatic breast cancer and can be used as an anti-oxidation, anti-cancer agent that does not compromise the therapeutic efficacy of radiotherapy [96,97,98,99]. 

Meanwhile, coumestrol (COU), found in various beans, leafy greens, and sunflower seeds, has the strongest relative binding affinity for ERβ at 140. Dietary COU intake has been shown to decrease ERα mRNA and protein levels like GEN, indicating possible usage as an anti-cancer therapeutic [98,99,100,101].

In summary, the literature suggests that phytoestrogens can act as anti-cancer agents by competing with endogenous estrogens, particularly with differences in relative binding to different ERs. While outcomes vary with tissue location and cancer types, the physiologically relevant research into phytoestrogens seems promising and will help to better understand the biological activities of these plant-produced estrogen mimics.

## 3. Advanced Methodology in Studying the Biological Effects of Xenoestrogens and Phytoestrogens

While population-based studies have defined correlations with environmental estrogen exposure and cancer, and cell/molecular studies have revealed some mechanisms for these effects, several novels approach to investigating the estrogen-cancer link are revealing more sophisticated insights [102,103]. Recently, nonbiased “multi-omic” approaches, including genomics, transcriptomics, proteomics, and metabolomics, have been widely applied to reveal the mechanisms at the molecular level [104,105,106]. In addition to these in vitro studies, the use of in vivo rodent models has been useful for studying the phenotypic changes and mechanisms of exogenous estrogen exposure. The main advantage of in vivo models is the ability to test a given chemical in a more relevant setting to humans so that the results can be more reasonably extrapolated at the tissue and systemic levels [107]. A better understanding of the “pros and cons” of each methodology and proper exploitation, or a transdisciplinary approach, will better progress the study of the causal relationship between exogenous estrogen exposure and human cancers. The Breast Cancer and the Environment Research Program (BCERP), funded by the US government, is a multi-institutional, multi-disciplinary group of teams of laboratory-based scientists, epidemiologists, social scientists, and clinicians with various specialties and from different perspectives. Our group is included in this program (https://bcerp.org/ (accessed on 16 May 2021). In this section, we describe our experience within the program context in developing new ways to study the biological effects of xenoestrogens and phytoestrogens in breast cancer. 

### 3.1. In Vitro Models with Cultured Cells

In vitro models with cultured cells are powerful tools for screening and identifying the estrogenic activity of chemicals existing in the living environment [102,103]. E-screen is a cell-proliferation assay that uses estrogen-dependent cancer cell lines to elucidate the estrogenic effects of these environmental chemicals [108,109]. Additionally, many gene reporter assays have been developed using human cancer cell lines transfected with reporter genes to assess whether a given compound could induce ER-mediated gene expressions [110,111]. We previously generated a model cell line by stable transfection with the estrogen-responsive element (ERE)-driven luciferase reporter into an aromatase-overexpressing MCF7 human breast cancer cell line, named MCF7aro-ERE [112]. We successfully developed the AroER Tri-screen assay system with MCF7aro-ERE, which is an improved model that is suitable for a high-throughput screening system [113]. AroER Tri-screen assay shows luciferase activity when estrogen-bound ERs induce gene expression by binding to the ERE promoter region. The AroER Tri-screen is a robust bioanalytical assay that has a high signal-to-background ratio, enabling the application of a high-throughput format of up to 1536 wells in a single experiment. In addition, this system is a multiplex assay, used not only for the screening of ER-agonistic chemicals but also for screening of ER-antagonistic or an aromatase inhibitor (AI)-like compound [114]. The AroER Tri-screen system has been adopted into a collaborative project called “Tox21” (https://tox21.gov/ (accessed on 16 May 2021)), which aims to develop target-specific, mechanism-based, and biologically relevant in vitro assays to screen for health-hazard chemicals. In this Tox21 program, we have utilized AroER Tri-screen to test a library of 10,000 compounds for anti-aromatase activity. The screen revealed 10 novel inhibitors. For example, imazalil, a widely used agricultural fungicide, showed irreversible and long-lasting anti-aromatase activity [115]. These high-throughput screening assays remain important for exploring exogenous estrogens among a large collection of chemicals, in newly developed consumer products, industrial chemicals, and drugs, as well as in unknown phytoestrogens from foods or plants, in a timely and reproducibly manner.

### 3.2. In Vivo Models with Rodents

In vivo models with rodents have been useful for studying the phenotypic changes and mechanisms of exogenous estrogen exposure. These models include carcinogenesis models and therapeutic models, with the former consisting of healthy or genetically engineered mice upon long-term exposure and the latter using established tumor xenografts. Conventional xenograft models using human cell lines or spontaneous mouse tumors have the limitations that they do not necessarily recapitulate the nature of original human cancers, leading to a lack of predictive value of the results in a clinical setting [116,117]. More specific and cancer-relevant PDX models, generated by the direct implantation of tumor fragments from human patients into immune-deficient mice, are increasingly being utilized for translational cancer research because they have been proven to maintain many of the biological properties of human cancers, such as genetic features, histology, and tumor cell population heterogeneity [118,119,120]. Many studies have reported that the response to treatments in PDX models correlates well with the results of treatment in the patients whose tumors supplied the PDX cancer. Therefore, PDX models provide a suitable option for studying the effects of exogenous estrogens on human cancers [121,122,123]. For example, the xenoestrogen methylparaben was shown to promote tumor growth and stem-like features using an ER^+^ breast cancer PDX model [124]. Our group has also recently performed bulk RNA-seq analysis on an ER^+^ breast cancer PDX treated with PBDEs, concluding that PBDEs induced the expression of estrogen-responsive genes, especially those related to cell proliferation [125]. Other groups also reported the effect of GEN [126,127] and DES [128] on prostate cancer PDX models. Additionally, another study investigated the potential chemo-enhancing effects and mechanisms of GEN and its analog AXP107-11, which showed an improved bioavailability of AXP107-11 for clinical use compared to GEN [129]. These findings suggest that PDX models would help further the understanding of the biological effects of exogenous estrogens as relevant models of human cancers. 

In addition to its advantage in mimicking the natural situation of tumor development, PDX models include all the cells in the surrounding tissues, rather than just the cancer cells, enabling the assessment of the biological effects on the whole population in a tissue and the specific cell-to-cell interactions [130]. Furthermore, we can observe various phenotypical changes, such as tumor invasion, metastasis, or immunomodulation [131], beyond simple cell proliferation or gene expression that can easily be observed in in vitro models. In contrast, there are also several disadvantages of in vivo models against in vitro models. First, animal models are often time- and cost-consuming, which limits their usefulness for exploratory studies as discussed in the in vitro screening assays [107]. Second, the experimental dosage used in animal studies is often much higher than typical human exposures, making the extrapolation to the human situation problematic [132]. In the real situation in human tumor development, exposure to low doses of xenoestrogens may result in subtle effects that accumulate over time. These are difficult to observe in animal studies. In addition, ethical considerations of animal use must be considered, especially when testing compounds in the cosmetic or consumer product industry [133].

### 3.3. Single-Cell RNA-Sequencing (scRNA-seq)

Tumor development and progression are widely recognized as complicated processes in which tumor cells, and many other contributors such as fibroblasts, immune cells, and other stromal cells from the tumor microenvironment, play distinct roles by their interactions with one another. Thus, the heterogeneity of cell populations within tissues of interest has been one of the major limitations of previous, especially with in vitro models. Additionally, even in the in vivo models, it is sometimes a challenge to capture the effect of estrogenic compounds in each type of cell, especially when those cells are too minor to cause apparent phenotypic changes. The recent development of scRNA-seq provides transcriptomic information at a single-cell resolution, enabling the ability to profile each isolated cell’s characteristics from a given tissue or organ [134,135]. This unprecedented capability of scRNA-seq technology allows us to capture subtle changes caused by xenoestrogens/phytoestrogens and their targeted cells, not only in the tumor cells of interest but also in the surrounding stromal cells (e.g., fibroblasts or immune cells), furthering the understanding of the potential interactions between these heterogeneous cell populations. Thus, this information can greatly help to reveal the mechanisms of cancer-initiating and/or promoting the effects of exogenous estrogens. 

We have demonstrated that this state-of-art technology can overcome some of the limitations of the pre-existing in vitro and in vivo models. We previously reported a study using scRNA-seq analysis on normal mouse mammary glands of a surgically menopaused mouse model treated with estrogen and PBDEs [136]. Our results suggest that PBDEs enhance estrogen-mediated mammary gland regrowth through the up-regulation of *Areg* expression in mammary epithelial cells, which in turn affects its cognate receptor, EGFR expressed on mammary fibroblasts and further modulates the recruitment of tumor-promoting M2 macrophages. These findings support the hypothesis that PBDE exposure with estrogen treatment increases the risk of breast cancer development during a critical period, menopause. ScRNA-seq analysis also provides fundamental insights into the regulatory activity of PBDEs on distinct populations in normal mammary glands in the presence of estrogen. Furthermore, we expanded our scRNA-seq analysis to study the effect of PBDEs on the differentiation of mammary epithelial cells by integrating human and mouse datasets from our and others’ studies, thereby constructing a mammary cell gene expression atlas [137]. One group utilized scRNA-seq technology, although not directly related to cancer research, to investigate the transcriptomic changes induced by a known xenoestrogen, di (2-Ethylhexyl) phthalate (DEHP), exposure. They revealed the reproductive toxicity of DEHP in murine germ cells and pre-granulosa cells at a single-cell level [138]. Although scRNA-seq has some limitations, such as technical noise from the cell preparation process, loss of spatial information, higher costs than other models, and requirement for freshly prepared samples [139,140,141], it serves as an excellent option for studying the complicated activity of xenoestrogens/phytoestrogens in heterogeneous cell populations of target tissues. 

## 4. Biological Activities and Mechanisms of Xenoestrogens and Phytoestrogens in Cancers

### 4.1. Effects of Xenoestrogens and Phytoestrogens on the Bioavailability and Formation of Endogenous Estrogens

Human sex hormone-binding globulin (hSHBG) is a high-affinity binding protein in the bloodstream for endogenous estrogens, modulating the bioactivity of estrogens by limiting their diffusion into target tissues and cells [142]. By binding to hSHBG, xenoestrogens and phytoestrogens could modulate the bioavailability of endogenous estrogens [143]. Meanwhile, extra-glandular tissues can also synthesize estrogens from adrenal dehydroepiandrosterone (DHEA) and androstenedione (4-dione) by steroidogenesis enzymes, such as aromatase and 3beta- and 17beta-hydroxysteroid dehydrogenases (3β-HSDs and 17β-HSDs) [103]. These exogenous estrogens can also disrupt extra-glandular estrogen formation via interruption of steroidogenesis enzymes (Figure 1). 

Xenoestrogens, such as BPA, NP, and monobutyl phthalate (MBP), have displayed a high binding affinity for hSHBG, with reversible and competitive binding activity for both testosterone and E2. Therefore, xenoestrogens may displace endogenous testosterone and E2 from hSHBG binding sites, leading to an increased level of free-form E2 in circulation. On the other hand, hSHBG may transport these xenoestrogens to target tissues and facilitate their diffusion into target cells [144]. Moreover, studies have found that xenoestrogens, such as BPA, exert their impacts on steroidogenesis by promoting aromatase expression in the adrenal cortex and ovaries; the increase of aromatase expression is responsible for the E2 increase [145,146]. This effect promotes the activation of ERα, which plays a pivotal role in the regulation of endocrine disorders such as cancer.

The flavonoid phytoestrogens, such as GEN and naringenin, have also been identified as hSHBG ligands [147]. Several studies in women have suggested a significant positive correlation between the intake of phytoestrogens and the concentration of plasma hSHBG [148]. Studies have also shown that the intake of phytoestrogens is negatively correlated with the plasma percentage of free-form E2 [149]. Such observations were further validated in large cross-sectional studies in postmenopausal women. Results have shown that phytoestrogen exposure is associated with lower plasma E2 in postmenopausal women and interacts with hSHBG levels and estrogen metabolism [150]. Dietary phytoestrogens suppress adrenal and ovarian 3β-HSDs and aromatase gene expression, therefore, decreasing estrogen formation [151]. Isoflavones have also been shown to exert inhibitory effects on 17β-HSD1 [152]. Amongst the phytoestrogens, isoflavones are the most potent inhibitors of aromatase [153]. Many phytoestrogens decrease the plasma estrogen levels, pointing towards a possible inhibitory effect in the regulation of E2 synthesis via suppressing the expression and activity of aromatase [154,155,156]. 

In summary, xenoestrogens and phytoestrogens may have distinct effects on the bioavailability and formation of endogenous estrogens. Xenoestrogens are more likely to displace endogenous E2 from hSHBG binding sites, enhance E2 formation by inducing steroidogenesis enzyme expression, such as aromatase, consequently promoting estrogenic responses in humans. Meanwhile, supplementation with phytoestrogens may lead to decreased plasma E2 levels via interaction with hSHBG levels and interruption of estrogen metabolism (Figure 1). 

### 4.2. Effects of Xenoestrogens and Phytoestrogens on Estrogen Receptor Activation and Signaling 

The variety of ERs reflects the diversity of receptor mechanisms involved with xenoestrogen and phytoestrogen effects on cells. This has relevance to the effects on these estrogenic molecules in cancer. There are two types of ERs: intracellular ERα and ERβ and membrane-associated mERs and GPER [157]. The intracellular ERα and ERβ belong to a group of nuclear receptors that act as ligand-activated transcription factors. They are also the primary receptors for both endogenous and exogenous estrogens. ERs are activated in four ways (Figure 2): (1) the classical genomic pathway where estrogens are bound to ERs that will activate the transcription of target genes, (2) the non-classical genomic pathway involving ER interactions with other transcription factors such as activator protein 1 (AP-1), including c-Fos, c-Jun, and c-myc, (3) the E2-independent pathway which activates ERs through phosphorylation induced by growth factor (EGFR/IGFR/Her2/3) signaling cascades [16], and (4) the non-genomic pathway involving membrane-associated ERs such as mERs and GPER [157].

The activation of ER signaling pathways plays a vital role in the malignant progression of multiple cancers by comprehensively regulating downstream genes. ERα activation has been shown to exert pro-oncogenic responses while ERβ activation has been shown to exert tumor-suppressive responses. These differences play a large role in the overall prognosis of patients with cancers [158,159]. Most xenoestrogens, including PBDE congeners and BPA, are agonists of both ERα and ERβ. They can mimic endogenous estrogens by interacting with ERα and ERβ, leading to phenotypic changes in cell proliferation, apoptosis, or migration [160]. These cellular changes contribute to the development and progression of hormone-related cancers in the breast, ovaries, and prostate [161]. In recent studies, many lines of evidence have also revealed that BPA exerts its function via activation of human estrogen-related receptor gamma (ERRγ), which behaves as a constitutive activator of transcription [162]. BPA preserves ERRγ’s basal constitutive activity and protects the selective ER modulator, 4-hydroxytamoxifen from its deactivation of ERRγ. This provides possible support that BPA exposure from the environment may potentially induce tamoxifen resistance to breast cancer treatment [163].

However, according to the literature, phytoestrogens such as GEN, DAI, and COU, along with others, exert a much stronger binding affinity for ERβ than for ERα [164]. For instance, GEN is a full ERβ agonist and, to a much lesser extent (~20-fold) of ERα [25]. Therefore, it is believed that the anti-cancer effects of these phytoestrogens may be due to their interactions with ERβ. ERβ in MCF7 breast cancer cells increases the anti-cancer efficacy of GEN by affecting cell cycle transition [165]. Several studies have also reported that GEN inhibits the cell cycle division of human prostate cancer cells via ERβ activation [166]. It is worth noting that the ERα and ERβ may mediate distinct biological effects in many tissues such as the mammary glands, prostate, lungs, and intestine in both males and females. Therefore, the ERα/ERβ ratio is an important factor to consider when predicting the response of cancer cells to phytoestrogen treatment [167]. In addition to ERα/ERβ, flavone and isoflavone phytoestrogens were also ligands of estrogen-related receptors (ERRα/ERRγ). These phytoestrogens induced the activity of ERRs [168]. 

Although the majority of xenoestrogens/phytoestrogens are believed to exert their biological effects through ERα and ERβ modulation, many of these compounds also activate ERs via a non-genomic pathway which involves mERs and GPER [169,170]. Especially in cancer cells, exogenous estrogens can bind to mERs and/or GPER and activate signaling cascades (Akt, MAPK) through the recruitment of protein kinases (Src and PI3K), therefore mediating rapid transcriptional events [171]. Xenoestrogens/phytoestrogens also activate ERs through phosphorylation induced by growth factor signaling cascades, for instance, the crosstalk between the EGFR/IGFR/Her2/3 growth factor signaling pathways [172]. Phytoestrogens such as GEN can inhibit MCF7 cell proliferation by inactivating the IGF-1R-PI3K/Akt pathway and decreasing the Bcl-2/Bax mRNA and protein expressions [173]. However, xenoestrogens such as BPA and NP can mediate EGFR signaling activation in lung cancer, causing an increase in proliferation, clonogenic growth, and tumor spheroid formation [174]. 

In conclusion, xenoestrogens/phytoestrogens mimic endogenous estrogens by binding to and activating different types of ERs (ERα, ERβ, mER, and GPER), orphan nuclear receptors (such as ERRα and ERRγ), and cross-talking with many other membrane-associated growth factor receptors (Figure 2). Xenoestrogens/phytoestrogens could act as either an agonist or display antagonistic activity, when endogenous estrogen is present, in a tissue-selective and spatiotemporal manner in human cancers. 

### 4.3. Effect of Xenoestrogens/Phytoestrogens on Activation of AhR/PPARγ/ROS Pathways

AhR (aryl hydrocarbon receptor), binds many types of molecules, including phytoestrogens and xenoestrogens, entering the nucleus and acting as a transcription factor. Because it is also activated by many environmental pollutants. AhR has been called a “xenobiotic sensor”. A major action for activated AhR is enhanced transcription of genes encoding CYPs, some of which are involved in estrogen biosynthesis [175]. In addition, there are interactions between the AhR and ER signaling pathways, with AhR agonists having anti-estrogenic activities. The mechanisms involve (1) AhR competes with ERs for promoter binding, leading to inhibition of ERs signaling, (2) activation of AhR signaling regulates E2 production by controlling the gene expression of CYP19, and (3) activation of AhR signaling ubiquitinates ERs for degradation via the proteasome, leading to inhibition of ER signaling (Figure 2) [176]. Phytoestrogens from soy (GEN, DAI, and S-equol) and licorice roots (liquidities) negatively regulate ERs activation via binding to AhR [177]. However, xenoestrogens, like PCBs and BPA, act selectively through AhR xenobiotic response element (XRE) and enhance AhR target-gene expression, including CYP19, therefore increasing endogenous E2 production [178]. Both ERs and AhR should be considered mediators of the biology, toxicology, and pharmacology of exogenous estrogens. 

In addition to the AhR signaling pathway, PPARs can also be activated by exogenous estrogens. PPARs belong to a family of nuclear receptors that act as transcription factors. They have comprehensive impacts on diabetes, adipocyte differentiation, inflammation, and cancer [179]. PPARα stimulation appears to inhibit the proliferation of human colon cancer cell lines and reduce polyp formation in the mouse model of familial adenomatous. PPARβ (also referred to as PPARδ) has been described in the regulation of keratinocyte differentiation, apoptosis, inflammation, and wound healing. PPARγ not only controls the expression of genes involved in differentiation but also negatively regulates the cell cycle [180]. BPA analogs have been reported to be ligands of ERs and PPARs; the greater their capability to activate PPARγ, the weaker their estrogenic potential is [181]. Meanwhile, the activation of PPARγ by GEN can down-regulate the transcriptional activity of ERα or ERβ in breast cancer cells [180,182]. Xenoestrogens/phytoestrogens concurrently activate ERs and PPARs, which may exert opposite biological effects. As a result, the balance between activated ERs and PPARs determines the biological effects of exogenous estrogens and estrogen-like mimics on cells and tissues (Figure 2). 

In addition to regulating cell functions through interactions with estrogen signaling, xenoestrogens and phytoestrogens can affect cells through oxidative stress signaling by generating reactive oxygen species (ROS) within healthy cells or cancer cells (Figure 2). Oxidative stress-mediated signaling is a double-edged sword in cancer cell behavior. Oxidative stresses are suggested to play important roles in estrogen-induced breast carcinogenesis [183]. There is growing evidence that the induction of ROS by BPA may contribute significantly to its genomic toxicity and carcinogenic potential [184,185]. On the contrary, many chemotherapeutic strategies are designed to significantly increase cellular ROS levels, leading to tumor cell apoptosis [186]. As noted above, the phytoestrogen COU is a potential chemotherapeutic agent for breast cancer. Evidence indicates that COU acts by inducing intracellular ROS, coupled with DNA fragmentation, up-regulation of p53/p21, cell cycle arrest, mitochondrial membrane depolarization, and caspases 9/3 activation [187].

### 4.4. Effects of Xenoestrogens/Phytoestrogens on Modulating the Hallmarks of Cancer 

Xenoestrogens/phytoestrogens primarily modulate the hallmarks of cancer cells by inappropriately activating ERs, cross-talking with membrane-associated growth factor receptors (EGFR/IGFR/Her2/3), and many other nuclear receptors (AhR/PPARs/ERRα/γ). In the presence of active signaling, the hallmarks acquired by cancer cells are modulated and linked to cell cycle and checkpoint disruption, metabolic rewiring, regulation of apoptosis, and redox homeostasis [188]. In addition to cancer cells, tumors exhibit another dimension of complexity by recruiting heterogeneous cell types and creating a “tumor microenvironment”. These cells include tumor-infiltrating immune cells, cancer-associated fibroblasts (CAFs), cancer-associated adipocytes (CAAs), and more [189]. The impact of exogenous estrogens on tumor-associated cells is significant (Figure 3). 

Immune modulation has been recognized as an emerging hallmark feature of cancer, including tumor-promoting inflammation and evading immune destruction [190]. Tumor-promoting inflammation is mainly characterized by the activation of innate immune cells, such as monocytes, macrophages, and natural killer cells (NK) often within the tumor environment. The innate immune cells subsequently increase the release of pro-inflammatory mediators such as TNF-α, IL-6, and IL-1β, which in turn stimulates the production of cyclooxygenase products and promote cancer progression. Evading immune destruction involves mechanisms of the adaptive immune cells (cytotoxic T cells, T helper cells, and B cells) by modulating certain immune checkpoint pathways [191]. The receptors (ERs, PPARs, and AhR) that bind xenoestrogens/phytoestrogens are present in lymphocytes, macrophages, neutrophils, and other immune cells [192]. Exposure to xenoestrogens increases the incidence of inflammation by activation of AhR and PPARγ [193]. Considerable research has found that GEN, a natural PPARγ agonist found in soy foods, exhibits anti-inflammatory activities via TNFα-induced NF-κB-dependent IL-6 gene expression by interfering with the mitogen- and stress-activated protein kinase 1 activation pathways [194,195]. 

However, the mechanisms of xenoestrogens/phytoestrogens via ER pathways in human immune cells have not been well studied. Their molecular mechanisms are based on interactions with ERα and ERβ, as well as with membrane associated GPER [196]. The expression of ERs in immune cells has various levels. For example, human CD4^+^ T cells and macrophages express higher levels of ERα than ERβ [197]. Xenoestrogens (BPA, DEHP, and PBDE) tend to stimulate M2-like tumor-associated macrophage (TAM) polarization and migration via simultaneously activating ERα or ERβ signaling pathways [198]. ERβ is involved in mediating estrogen action on NK cell activity [199]. Isoflavones such as GEN decrease IL-12/IL-18-induced IFN-γ production in NK cells without altering NK cell cytotoxicity. The regulation of NK cells via ERβ may be linked as a benefit of the anti-inflammatory and anti-cancer process of phytoestrogens [200]. 

Cancer-associated fibroblasts (CAFs) and cancer-associated adipocytes (CAAs) within the tumor environment have recently been implicated in important aspects of epithelial cancer biology such as neoplastic progression, tumor growth, angiogenesis, and metastasis. CAAs from adipose tissue may contribute to breast cancer development and progression by altering neighboring epithelial cell behavior and phenotype through paracrine signaling [201]. Many xenoestrogens have been shown to cause obesity in animals at low-level exposures during critical periods of development. More specifically, DES and BPA have been implicated as environmental chemicals that increase fat accumulation by increasing the number of adipocytes, storage of fat within adipocytes, and facilitating obesity [202]. BPA is reported to exert estrogen-like activity on CAFs, particularly through the GPER. BPA induces the expression of GPER target genes, c-*FOS*, *EGR-1*, and *CTGF*, through the GPER/EGFR/ERK transduction pathway in CAFs, leading to their growth and migration in breast cancer [203]. On the contrary, dietary exposure to soy foods is associated with lower mammary tumor risk and a reduction in body weight and adiposity in human and rodent breast cancer models [204]. GEN has been shown to lower mammary adiposity and increase mammary tumor suppressor expressions, such as PTEN and E-cadherin, in female mice. These modulations mediate through ERβ and PPARγ by promoting the differentiation of stromal fibroblasts into mature adipocytes [205]. These results suggest that the direct regulation of mammary adiposity by GEN could be useful for breast cancer prevention. 

The effects of xenoestrogens and phytoestrogens on the tumor microenvironment are challenging to study. Traditional animal models that use homogeneous cancer cells do not mimic the actual dynamic, multicellular environment of a human tumor. Therefore, advanced research models, such as PDXs and scRNA-seq technology, allow scientists to capture changes caused by xenoestrogens/phytoestrogens in both cancer cells and the surrounding stromal cells, ultimately improving the understanding of the interactions among these heterogeneous cell populations.

### 4.5. Effects of Xenoestrogens/Phytoestrogens Determines on Critical Timing of Exposure 

Endogenous estrogen flux has been linked to increased breast cancer risk through critical estrogen exposure during certain events and time points during the life cycle such as nulliparity, older age at first birth, early menarche, and late menopause [206]. By the same principle, there is a consensus that the influence of environmental estrogens on breast cancer risk may be greater during certain WOS in a woman’s life. WOS are key life stages in which mammary glands undergo anatomical or molecular transformations and are most vulnerable to environmental exposures. The risk of breast cancer development increases if xenoestrogen/phytoestrogen exposure occurs during WOS, including prenatal development, puberty, pregnancy, and menopausal transition [23]. Exposures to xenoestrogens such as BPA and triclosan can change the timing of puberty and cause early breast development [207]. Menopause is a critical WOS because of its hypersensitivity to endocrine-disrupting chemicals due to the decline of endogenous estrogen [208]. Studies from our group have discovered that PBDEs, the flame retardants in household products, enhance E2-mediated regrowth of mammary glands, augment E2-facilitated gene expression, and modulate immune regulation, thus increasing the risk of developing breast cancer [136,137,138,139]. Importantly, like the WOS in female breast cancer, there appears to be a heightened sensitivity of the prostate to these exogenous estrogens during the critical developmental windows, such as in utero, the neonatal period and puberty. Thus, it is suggested that infants and children may be considered a highly susceptible population for exogenous estrogenic exposure with increased prostate cancer risk with aging [209].

The biological effects of phytoestrogens on breast cancer have also been linked to age and critical time points in a woman’s life [210]. In premenopausal women, who are at high risk for early breast cancer, dietary isoflavone intake has been associated to increase breast cell cancer risk by promoting cancer cell growth. However, isoflavone intake appears to have a protective impact on later breast cancer recurrence and mortality among postmenopausal breast cancer patients [211]. On the other hand, some phytoestrogens appear to reduce breast cancer throughout life. Asian diets, with abundant soy products, include phytoestrogens that appear to be chemo-preventive for breast cancer in Asian women, who consume more soy than women who consume a Western diet [212]. However, the relevant research on phytoestrogens in breast cancer is complicated, inconsistent, and inconclusive [213]. 

In addition to their influences on the etiology of hormone-related cancers, the impacts of xenoestrogens/phytoestrogens on reproductive health are manifested and determined based on the critical timing of exposure. Early life exposure alters the development of both female and male reproductive systems. The greatest risk may be during the prenatal (fetus) and early postnatal (infant) developmental windows when the organs are forming and developing [214]. Xenoestrogenic/phytoestrogenic exposure in young children may lead to early activation or interference with the hypothalamic-pituitary-gonadal (HPG) axis and therefore contribute to the early onset of puberty [215]. In adults, BPA, phthalates, pesticides, etc. have been shown to decrease the number of primordial follicles in female ovaries [216] and decrease the number and motility of sperm in male semen [217]. 

Xenoestrogens/phytoestrogens can also influence non-reproductive tissues and are involved in the etiology of disorders including obesity and diabetes mellitus [218], cardiovascular and respiratory disease [219], neurological effects [220], and thyroid disease [1]. It is not surprising that the influences of xenoestrogens/phytoestrogens on these disorders are also associated to the critical timing of exposure. For instance, BPA exposure in women of reproductive age, including pregnant women, has been linked to an increased risk of insulin resistance and type 2 diabetes [221]. For women with BPA exposure during pregnancy, their offspring have a greater chance of having a higher diastolic blood pressure at an early age [222]. There is also correlative evidence suggesting that xenoestrogenic exposure during pregnancy, breastfeeding, and early in childhood may interfere with normal brain development, either directly or indirectly, by disrupting the thyroid hormone signaling axis [223]. More specifically, current literature has shown that many xenoestrogens disrupt thyroid functions through their influence on the thyroidal hormones, triiodothyronine (T3) and thyroxine (T4). These disruptions can lead to their indirect downstream effects in various developmental windows or human life stages. For instance, GEN and PCBs can disrupt thyroid transport proteins, resulting in hormone fluctuations that have been associated with impaired neurodevelopment in offspring [224], whereas PBDE exposure has been associated with hypothyroidism [225]. 

## 5. Application of Phytoestrogens in the Prevention or Treatment of Cancers: Evidence from Clinical Trials

Phytoestrogens such as soy isoflavones DAI, GEN, and glycitein are dietary components that are thought to reduce the incidence and severity of various cancers [226]. The assumed benefits of this soy diet have led to numerous clinical studies on phytoestrogen efficacies to determine a suitable amount for human consumption without any adverse effects. Additionally, clinical studies of phytoestrogens combined with cancer treatments are underway to observe if there is a synergistic effect to treat cancer. Here, we have reviewed 18 clinical trials [61,227,228,229,230,231,232,233,234,235,236,237,238,239,240,241,242,243,244], conducted between 2002 to present, focused on breast cancer (seven trials) [227,228,229,230,231,232,233,234], prostate cancer (eight trials) [61,235,236,237,238,239,240,241], endometrial cancer (two trials) [242,243], and colon cancer (two trials) [240,244], combined with two categories of phytoestrogens treatments: fruits/whole grains/seeds such as resveratrol and curcumin (eight trials) and soy isoflavones such as GEN (10 trials) (Table 3 and Appendix A). 

Of the 18 trials, in terms of safety, four trials have shown that phytoestrogens are well-tolerated, safe to use, and/or have no major safety concerns. One trial studies prostate and colon cancer in phase 1 (NCT02138955) while two trials studies breast cancer in phase 2 (Nr 5592-17-02-23) and 3 (NCT00513916). In terms of the efficacy, seven trials showed little or no evidence that phytoestrogens were antagonistic to breast cancer (four trials, NCT01219075, NCT00597532, NCT00612560, and NCT00290758), or prostate cancer (two trials, NCT00255125 and NCT02724618), or endometrial cancer (one trial, NCT00118846). Meanwhile, a total of six clinical trials have shown no significant differences between the treatment and placebo groups, including two breast cancer trials (NCT00290758, NCT00597532), three prostate cancer trials (NCT01009736, NCT01917890, NCT02724618), and one endometrial cancer trial (NCT00118846). Additionally, four clinical trials stated that the conclusions were not statistically significant, including one breast cancer trial (NCT00597532), two prostate cancer trials (NCT00255125, NCT0191789), and one endometrial cancer trial (NCT00118846). Lastly, five clinical trials consisting of two breast cancer studies (University of North Dakota School of Medicine and Health Sciences and NCT00513916), one prostate cancer study (NCT00546039), one endometrial cancer study (NCT02017353, phase 2), and one colon cancer study (NCT00256334, phase 1) suggested the need for larger and/or longer studies.

While the clinical rials of phytoestrogens noted above gave few promising results, combinations of a phytoestrogen with an established chemotherapy drug may be a more promising approach. For example, patients receiving CUR and Paclitaxel to treat metastatic breast cancers had a greater objective response rate (*p* < 0.05 16 weeks after starting treatment, and *p* < 0.01 after completed treatment) compared to patients receiving Paclitaxel alone (Ministry of Health Republic of Armenia Registration No.: Nr 5592-17-02-23). Moreover, some men observed a slow rise of serum PSA after consuming 141 mg of isoflavones per day (NCT00596895). This prostate cancer trial has also shown that GEN may have an inhibitory effect on androgen-related biomarkers and supports GEN as a chemo-preventive agent in prostate cancer (NCT00546039). 

While tumor response has been used to evaluate the effectiveness of phytoestrogens in cancer treatment, more recent clinical trials have added gene expression analysis. Phytoestrogens alter cancer-related gene expression profiles in breast cancer (NCT00597532, NCT00290758, and University of North Dakota School of Medicine and Health Sciences trail), prostate cancer (NCT00546039), endometrial cancer (NCT02017353), and colon cancer (NCT00256334). More interestingly, some of the trials have shown that phytoestrogens are altering the cancer-related gene expression profiles [227,233,238,243,244]. Under the concept of personalized medicine, gene expression analyses could be an alternative and cost-effective way to predict the effectiveness of phytoestrogens in cancer prevention and treatments. However, a larger number of clinical trial participants and more studies of phytoestrogens and their impact on cancers are still needed to better define their anti-cancer potentials.

## 6. Future Directions and Conclusions

According to Global Cancer Statistics 2020, the burden of cancer incidence and mortality is rapidly growing worldwide. The epidemiological features of cancer reflect both the aging and growth of the population, as well as the changes in the prevalence and distribution of the main cancer risk factors, several of which are particularly associated with the environment [245,246]. Exogenous estrogens, such as synthetic industrial estrogenic compounds (xenoestrogens) and estrogenic molecules from plants (phytoestrogens), are a group of environmental factors that potentially cause various cancers through their interactions with cellular signaling processes involving estrogen signaling pathways. 

Current knowledge of environmental health, oncology, and epidemiology gives new insight into the etiology of human cancers because of the gene-environmental interactions [247]. However, available epidemiological assessments of the risk of human cancers, which are multifactorial and multistage diseases, do not reflect the complex interactions between the biology of humans and/or their chemical exposure, and any consequent adverse health effects [248]. Moreover, models for the risk assessment of cancers are often based on single-agent causality. Such approaches may miss the possibility of a relationship with exposure to multiple hazardous compounds [249]. For this reason, the effects of the mixture of xenoestrogens/phytoestrogens have not been adequately addressed. While in vitro models with cultured cancer cells provide an advantageous method to interpret the single-agent causality of exposure and disease. However, these models also fail to consider a multifactorial analysis to explore the causal relationship between exposure and cancer development/progression. A novel approach to investigate the complexity of cancer with advanced modes and emerging techniques will be helpful to interpret measurable environmental and biological parameters simultaneously. These emerging approaches include in vivo models with rodents, PDX models, multi-omics-based unbiased analyses, and single-cell analyses [250,251]. Using multidisciplinary approaches, the etiology of human cancer can be more thoroughly investigated. 

The Breast Cancer and the Environment Program (BCERP), launched by the US National Institute of Environmental Health Science (NIEHS) and National Cancer Institute (NCI), is a representative multidisciplinary research program that explores the environmental factors that may contribute to breast cancer (https://bcerp.org/ (accessed on 16 May 2021)). The BCERP involves a network of lab-based biologists, clinicians, epidemiologists, and community partners to examine the effects of environmental exposures that may predispose a woman to breast cancer throughout her life. Our team is a member of this project. By taking advantage of the PDX-breast cancer model and scRNA-seq analysis in surgically menopausal (ovariectomized/OVX) mouse models, our group has identified the response to exposure to the xenoestrogen PBDE by various types of cells within mammary tumors and normal breast tissue [125]. At the single-cell level, by integrating mouse and human datasets, we also describe the landscape of transcriptional changes in mammary glands upon endogenous and PBDE at different WOS in a woman’s life [136,137,139]. Other key findings from the BCERP include (1) proteins produced by the developing mammary tissue may change after BPA exposure, which may alter the cell behavior in ways that contribute to breast cancer [252]. (2) DDT exposure during pregnancy may change the pattern of gene expression, leading to an increased chance of developing breast cancer in female offspring [253]. (3) The BCERP overarches a concept that the influence of environmental chemicals on breast cancer risk may be greater during certain WOS in a woman’s life, including prenatal development, puberty, pregnancy, and menopausal transition, during which the mammary glands undergo anatomical and functional transformations. Therefore, environmental hormones (e.g., endocrine-disrupting chemicals/EDC), and certain therapeutics (e.g., prescribed for the coexisting medical conditions or in the form of the hormone replacement therapy) can influence breast cancer risk, development, or outcome [23]. WOS is different from the well-known concept of “Sensitive Windows of Development”, which is referred to the period of fetal development and childhood when hormones regulate the formation and maturation of organs. Therefore, early-life exposures have been linked to developmental abnormalities and may increase the risk for a variety of diseases later in life [214]. (4) Finally, data from the BCERP have described the biological activities and molecular mechanisms of xenoestrogens on mammary gland biology and neoplasia, providing a scientific consensus with an integrated source of information and technology, of the development, function, and pathology of the mammary gland upon xenoestrogen exposure (https://bcerp.org/ (accessed on 16 May 2021)). 

In addition to female breast cancer, there is increasing evidence from both epidemiology studies and animal models that environmental exposure to exogenous estrogens may influence the development or progression of prostate cancer, by interfering with estrogen signaling, either through interacting with ERs or by influencing steroid metabolism and altering estrogen levels within the body [254]. In humans, epidemiological evidence links specific pesticides such as the banned but still environmentally present PCBs exposures to elevated prostate cancer risk [255]. Studies in animal models also show augmentation of prostate carcinogenesis with several other environmental estrogenic compounds including BPA [256]. Recently, endogenous and exogenous estrogens have also been postulated as a contributor to non-classical hormone-related tumors, including lung cancer [257], colorectal cancer [258], and gastric cancer [259]. For instance, the etiology of lung cancer is mainly related to environmental exposure such as cigarette smoking and airborne genotoxic carcinogens. However, even correcting for carcinogen exposure, there appears to be an increased risk for lung cancer in women as compared to men. This suggests that sex hormones may be involved with lung carcinogenesis [260]. Several agents commonly present in the living environment can have dual biological effects: acting as genotoxic/carcinogenic and hormonally active xenoestrogens. The dualism of these environmental chemicals may contribute to the development and progression of lung cancer [261]. However, there has been a lack of solid evidence to prove the causal relationships between exogenous estrogen exposure and the increased risk of non-classical hormonal-related cancers. 

Different from the xenoestrogens which are widely accepted as carcinogens, a wide range of beneficial effects of phytoestrogens on the cardiovascular, metabolic, and central nervous systems, as well as a reduction of cancer risk and postmenopausal symptoms, has been claimed [262]. The benefits of phytoestrogens such as the soy diet have led to numerous clinical studies on phytoestrogen efficacies to prevent or treat cancer [61,225,226,227,228,229,230,231,232,233,234,235,236,237,238,239,240,241,242]. However, there is also concern that phytoestrogens may act as endocrine disruptors that adversely affect health [212]. Thus, clinical trials are underway to evaluate the safety and efficacy of phytoestrogens with breast, prostate, endometrial, and colon cancer, and more. Our review has included many phases I and II trials that have indicated the safety of phytoestrogens in humans. Existing data generally supports the safety of small doses of purified phytoestrogen consumption as a medication for breast cancer [225,226,227,228,229,230,231,232]. However, for the entire general population, including women with benign breast disorders, those at risk for breast cancer, and even survivors of cancer, the prescription of phytoestrogens is still not recommended due to insufficient evidence [211]. Under the framework of personalized medicine, several clinical trials [225,231,236,241,242] have suggested that phytoestrogens have been shown to change the cancer-related gene expression profiles, providing a perspective that gene expression analyses may help to better predict the effectiveness of phytoestrogens in cancer prevention and treatments. Moving forward, continued research into phase II and III trials with larger participant cohorts and more studies into phytoestrogens are needed to fully elucidate their anti-cancer benefits.

In conclusion, exogenous estrogens, particularly xenoestrogens and phytoestrogens are an important contributor to the development and progression of cancers. Future studies on etiology of human cancers related to environmental exogenous estrogen exposure should focus on synthesizing various perspectives: (1) at the molecular and cellular level, looking at different types of ERs (ERα, ERβ, mER, and GPER) and cross-talk with other signaling pathways, (2) at the tissue level, considering the spatial heterogeneity of tissue composition and temporal heterogeneity of cancer progression, (3) at the systematic level, studying the exposure time at critical developmental windows, and (4) at the individual or population level, considering gene-environment interactions. Incorporated analysis of all the data in a clearly understood fashion allows for the modeling of prevention and therapy on an individual basis and the potential for developing new diagnostic biomarkers and drugs. Moreover, in the future, closer collaboration among oncology, systems biology, and environmental health may provide a significant qualitative and quantitative leap forward in the elucidation of human cancer etiology. The information gained from such collaborations could be applied in the introduction of preventive measures, personalized medicine, and more relevant public health intervention, ultimately, improving the knowledge and management of the complex environmental interactions underlying this life-threatening disease. 

## Figures and Tables

**Figure 1 ijms-22-08798-f001:**
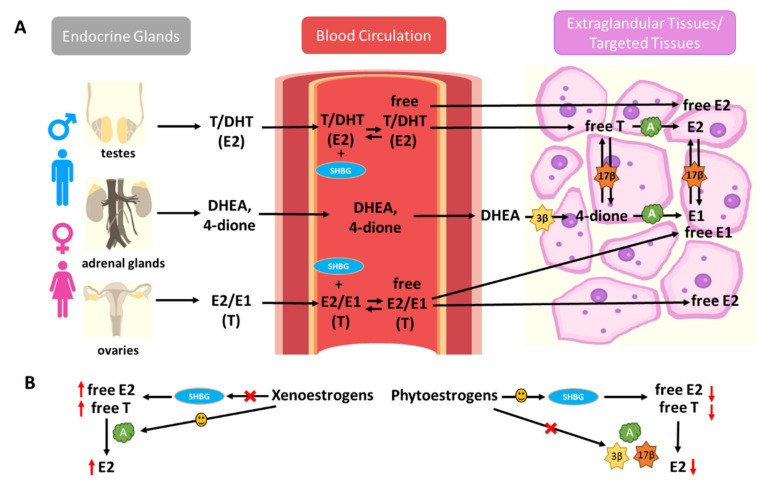
Xenoestrogens and phytoestrogens modify endogenous estrogen bioavailability and formation. (**A**) Endogenous estrogens are produced by endocrine glands (ovaries, testes, and adrenal glands) and transported to endocrine-responsive tissues through blood circulation. Human sex hormone-binding globulin (hSHBG) is a high-affinity binding protein in the bloodstream for endogenous estrogens, modulating the bioactivity of estrogens by limiting their diffusion into target tissues and cells. Extra-glandular tissues can also synthesize estrogens from adrenal dehydroepiandrosterone (DHEA) and androstenedione (4-dione) by steroidogenesis enzymes, such as aromatase (CYP19) and 3beta- and 17beta-hydroxysteroid dehydrogenases (3β-HSDs and 17β-HSDs). **(B**) Xenoestrogens and phytoestrogens can modify the bioavailability of circulating endogenous estrogens by interfering with hSHBG binding. Xenoestrogens can also disrupt extra-glandular estrogen formation via interruption of steroidogenesis enzymes (A, aromatase, 3β, 3β-HSDs, and 17β, 17β-HSDs). Xenoestrogens are more likely to displace endogenous E2 from hSHBG binding sites, enhance E2 formation by inducing the steroidogenesis enzyme expressions, such as aromatase, consequently promoting the estrogenic responses in humans. However, phytoestrogens may lead to a decrease in plasma E2 levels via interaction with hSHBG levels and interruption of estrogen metabolism.

**Figure 2 ijms-22-08798-f002:**
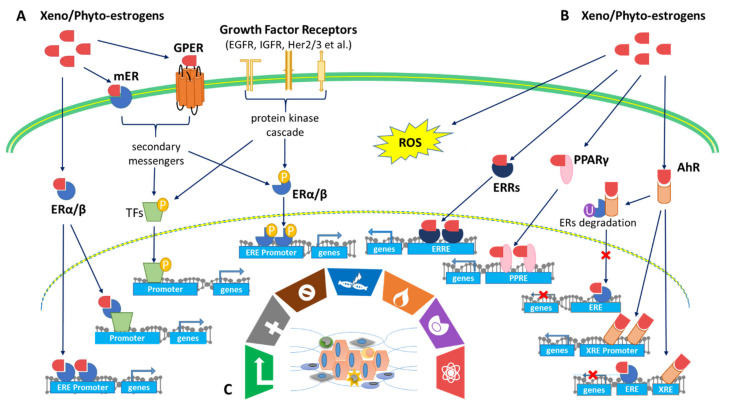
Xenoestrogens and phytoestrogens 
modulate multiple estrogen-mediated signaling pathways to shape the hallmarks 
of cancer. (**A**) Activation of estrogen receptor signaling. There are two 
types of ERs: intracellular ERα and ERβ and membrane-associated mERs and GPER [157]. ERs are activated in four manners: (1) the 
classical genomic pathway where estrogens are bound to ERs that will activate 
the transcription of target genes, (2) the non-classical genomic pathway 
involving ERs interactions with other transcription factors (TFs) such as 
activator protein 1 (AP-1), including c-Fos, c-Jun, c-myc, (3) the 
E2-independent pathway which activates ERs through phosphorylation induced by 
growth factors (EGFR/IGFR/Her2/3) signaling cascades [16], 
and (4) the non-genomic pathway involving membrane-associated ERs such as mERs and 
GPER. (**B**) Co-activation of AhR/PPARγ/ERRγ/ROS pathways. 
Xenoestrogens/phytoestrogens activate AhR signaling pathways and cross-talk 
with ER pathways: (1) AhR competes with ERs for promoter binding, leading to 
inhibition of ER signaling, (2) activation of AhR signaling regulates E2 
production by controlling the gene expression of CYP19, also known as 
aromatase, and (3) activation of AhR signaling ubiquitinates ERs for 
degradation via the proteasome, leading to inhibition of ER signaling. 
Xenoestrogens/phytoestrogens activate peroxisome proliferator-activated 
receptors (PPARs) and estrogen-related receptor gamma (ERRγ). 
Xenoestrogens/phytoestrogens could also induce oxidative stress-mediated 
signaling by generating reactive oxygen species (ROS). (**C**) Shaping the hallmarks 
of cancer. These features are linked to cell cycle and checkpoint disruption (
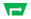
), cell apoptosis and death 
reprogramming (
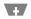
), growth suppressor evading (
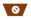
), genome instability and 
mutation (
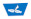
), tumor inflammation-promoting 
(
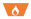
), immune response destruction 
(
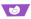
), redox homeostasis 
interrupting, and metabolic rewiring (
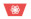
).

**Figure 3 ijms-22-08798-f003:**
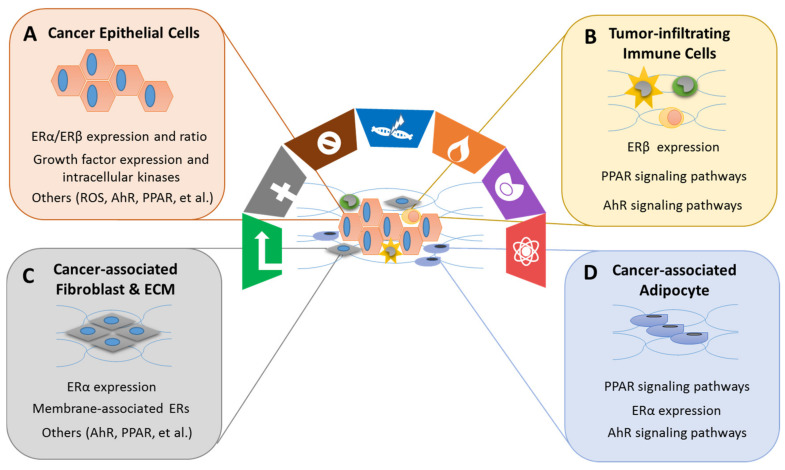
Xenoestrogens/phytoestrogens modulate the cancer cells and cancer-associated cells. Xenoestrogens/phytoestrogens modulate the cancer cells and cancer-associated cells by inappropriately activating ERs, cross-talking with membrane-associated growth factor receptors (EGFR/IGFR/Her2/3) [16], and many other transcriptional factors (AhR/PPARs/ERRα/γ). In the presence of active signaling, the hallmarks acquired by cancer have been are modulated and linked to (**A**) cancer epithelial cells, (**B**) tumor-infiltrating immune cells, (**C**) cancer-associated fibroblasts and extracellular matrix (ECM), and (**D**) cancer-associated adipocytes.

**Table 1 ijms-22-08798-t001:** Various xenoestrogens and their implications for cancer. Compilation of ten types of xenoestrogens and their sources, biological and experimental evidence from pre-clinical studies, and implications towards cancer. Relative binding affinities were adapted from Kuiper et al. [25], unless otherwise noted, with E2 set as 100.

Name	Source	Relative Binding Affinity		Biological Activity	Experimental Evidence	Public Health Implications	References
		ERα	ERβ				
bisphenol A (BPA)	chemical used to manufacture polycarbonate plastics, epoxy resins, and added to other plastics; found in food containers, utensils, dental sealants, protective coatings, flame retardants, water supply pipes	0.01	0.01	disrupts ER activity by mimicking, enhancing, or inhibiting endogenous estrogen; directly impacts intracellular signal transduction ↑ ER mRNA	↑ hyperplastic ducts ↑ ER+ cells ↑ PR+ cells ↑ cell proliferation Phosphorylation of AKT and ERK ↑ prostate cancer cell proliferation Aberrant development of prostate and urethra ↑ prostate tumor size AR antagonist ↑ SHBG	increased risk of breast, prostate, and uterine cancer no risk for ovarian cancer	[26,27,28,29]
dichlorodiphenyltrichloroethane (DDT)	pesticide; used to combat malaria, typhus, and other insect-borne human diseases	0–0.01	0–0.02	estrogenic activity	Accumulates in adipose tissue Stimulates uterine proliferation and impairs normal follicle development Inhibits PKA activation Alters gene expression and hormone synthesis. Inhibit PGE2 levels in ovaries	increased breast cancer risk	[30,31,32]
polychlorinated biphenyls (PCBs)	used as flame retardants; found in electrical equipment, construction materials, coatings, textiles, furniture padding, etc.	0.01–3.4	<0.01–7.2	estrogenic/anti-estrogenic	↓ cell growth ↓ proliferation ↓ AR activity ↑ competitive inhibition to AR ↑ uterus weight	increased breast cancer risk for certain PCBs	[36,37,38]
polybrominated diphenyl ethers (PBDEs)	used as flame retardants; found in electrical equipment, construction materials, coatings, textiles, furniture padding, etc.	1.3–20 ^a^		estrogenic activity	↑ viability and proliferation of human breast, cervical, and ovarian cancer cells ↑ cell contact ↑ phosphorylation of PKCa and ERK1/2 proteins in tumor cells and in CHO cells	no clear association with breast cancer risk	[39,40,41,42,43]
diethylstilbestrol (DES)	used to prevent miscarriage, premature labor, and pregnancy complications	236	221	hydrophobic interactions; potent transcriptional activator through genomic signaling	↑ PI3 kinase signaling ↑ AKT phosphorylation ERRγ antagonist ↑ SRC1 ↑ SHBG ↓ LH, TSH, FSH, DHEA, testosterone, and E1	vaginal cancer risk	[44,45]
methoxychlor (DMDT)	used to protect pets, crops, and livestock from pests such as mosquitoes, cockroaches, and other insects	<0.01	<0.01	ERα agonist ERβ antagonist anti-estrogen in ovaries	Inhibit estrogen binding to ER ↓ serum progesterone ↑ uterotrophic activity Impairs overall fertility	increased ovarian cancer risk	[33]
ethinyl estradiol (EE2)	ovulation inhibitor; used in hormonal contraceptives	190 ^c^		↑ERRγ and RAGE expression primarily through Erα	↑ cell proliferation but not as much as E2↑	little/no breast cancer risk reduced risk for ovarian, endometrial, colorectal, and lymphatic/hematopoietic cancers	[46]
phthalates	found in soft plastics used as packaging materials	N/A ^d^	N/A ^d^	competitive binding with E2 for ER	↑ MCF7 cell proliferation and viability	increased breast cancer risk	[47,48]
nonylphenols	used in industrial processes and in consumer laundry detergents, personal hygiene, automotive, latex paints, and lawn care products	0.0032–0.037 ^c^		estrogen-like activity on ER+ breast cancer cells	↑ prostate epithelial cell proliferation ↓ prostate cell viability Promotes cytoplasm-nucleus Translocation of ERα, but not ERβ	increased breast cancer risk	[49,50]
parabens	used as preservatives in many foods such as beer, sauces, sodas, and cosmetics	0.011–0.11 ^b^	0.011–0.123 ^b^	ERRγ agonist	breast cancer cell proliferation ↑ tumor size Sulfotransferase inhibitor	increased breast cancer risk	[51,52]

^a^ values were adapted from Cao et al. [40] and include hydroxy PBDEs. ^b^ values were adapted from Golden et al. [52], which included data from Kuiper et al. [25]. ^c^ values adapted from Blair et al. [49] were obtained using a different methodology; use with caution when making comparisons. ^d^ specific relative binding affinity values were not found.

**Table 2 ijms-22-08798-t002:** Various phytoestrogens and their implications for cancer. Compilation of ten types of phytoestrogens and their sources, biological and experimental evidence from pre-clinical studies, and implications for cancer. Relative binding affinities were adapted from Kuiper et al. [25], unless otherwise noted, with E2 set as 100.

Name	Source	Relative Binding Affinity		Biological Activity	Experimental Evidence	Public Health Implications	References
		ERα	ERβ				
genistein (GEN)	soybeans and soy-containing products	4	87	[low]: estrogenic [high]: anti-estrogenic ↓ ERα protein/mRNa levels	↑ apoptosis ↑ cell cycle arrest ↑ demethylation of tumor suppressor genes Inhibits ovarian cancer cell migration, invasion, and proliferation ↓ phosphorylation of PI3K and GSK3b RTK inhibitor DNA topoisomerase II inhibitor ER+ cell proliferation ↓ tumor associated macrophage ↓ proliferation VEGF inhibitor (angiogenesis) ↓ breast CSCs ↑ cell adhesion ↓ migration/invasion	breast and prostate cancer preventative decreased ovarian cancer risk	[53,54,55,56,57,58]
daidzein (DAI)	soybeans	0.1	0.5	anti-estrogenic in organs expressing more ERα estrogenic in ERβ-presenting organs	↑ ERa expression/nuclear localization ↓ cell proliferation ↓ migration ↓ invasion Induces cell cycle arrest and apoptosis	endometrial cancer preventative	[59,60]
quercetin (QUE)	various fruits and vegetables such as apples, red grapes, onions, raspberries, honey, cherries, citrus fruits, green leafy vegetables, red wine, cappers, lovage, radish leaves, tea, cranberries, and peppers	0.01	0.04	estrogenic ↓ cytoplasmic ER levels ↑ tighter nuclear association to ER	↑ antiproliferative ↓ mammospheres in breast cancer cells ↓ breast CSC characteristics ↓ EMT Regulates B-catenin signaling, leading to EMT inhibition [low]: ↑proliferation ↑ migration ↑ invasion ↓ apoptosis [high]: ↓cell growth ↓ metastatic process ↑ cell cycle arrest ↓ tumor volume	anti-cancer for breast cancer	[54,57,62,63,64,65,66,67,68]
apigenin (APE)	fruits and vegetables such as parsley chamomile, celery, vine-spinach, artichoke, oregano, red wine, and beer	0.3	6	↓ ERα in uterus estrogenic/anti-estrogenic ↓ estradiol levels	[low]: ↑proliferation ↑ AKT phosphorylation ↑ invasion [high]: ↓proliferation ↓ AKT phosphorylation ↓ invasion ↑ apoptosis ↑ cell cycle arrest ↓ cell growth Inhibit MAPK	decreased breast, prostate, and ovarian cancer risk	[61,62,69,70,71,72,73,74]
resveratrol (RES)	Japanese knotweed grapes, wine, strawberries, and peanuts	6.11–11.2 ^a^	4.7–15.66 ^a^	ERRγ agonist	↑ breast cancer cell proliferation ↑ tumor size Sulfotransferase inhibitor	increased breast cancer risk	[75,76,77,78,79]
myricetin (MYR)	vegetables, fruits, nuts, berries, tea, and red wine	N/A ^c^	N/A ^c^	Competitive binding to ER ERα agonist	Inhibits prostate cancer cell growth, key enzymes involved in the initiation and progression of cancer ↓ migration ↓ invasion ↓ adhesion ↓ tumor nodules ↓ MMP2 and MMP9 protein expression ↑ apoptosis CK2 inhibitor	decreased breast and prostate cancer risk	[80,81,82,83,84,85]
kaempferol (KPF)	tea, broccoli, apples, strawberries, beans, bee pollen, cabbage, capers, cauliflower, chia seeds, chives, cumin, moringa leaves, endive, fennel, and garlic	0.1	3	estrogenic activity ERα-dependent transcriptional activation activity	↑ apoptosis ↓ cancer cell growth ↓ angiogenesis Preserve/protect cell viability ↓ migration ↓ MMP3 protein activity Inhibit VEGF release in breast cancer cells Reduced VEGF mRNA in ovarian cancer cells ↓ tumor growth/metastasis ↓ EMT ↑ cell cycle arrest Inhibits various cancer cell lines	decreased breast cancer risk	[86,87,88,89,90,91]
luteolin (LUT)	celery, peppermint, thyme, rosemary, oregano, artichoke, green pepper, and perilla leaf	N/A ^d^	N/A ^d^	Estrogenic	↑ cell cycle arrest ↑ apoptosis ↓ proliferation Inhibit MAPK, EGFR, VEGF ↓ PSA ↓ aromatase ↓ ERK and FAK phosphorylation	anti-cancer for breast and prostate endometrial cancer risk	[92]
curcumin (CUR)	derived from the plant *Curcuma longa*; turmeric	N/A ^b^	N/A ^b^	↓ ER expression	↓ EMT and migration ability ↓ breast CSC population ↓ nuclear translocation of B-catenin (slug transactivation; restored E-cadherin expression) ↑ apoptosis ↑ cell cycle arrest ↑ senescence ↓ p53 Inhibits proliferation, migration, invasion, angiogenesis, and metastasis in breast cancer cells Interferes with osteoblast formation in prostate cancer cell line	anti-cancer	[93,94,95,96]
coumestrol (COU)	plants such as soybeans, clover, alfalfa sprouts, sunflower seeds, spinach, legumes, chickpeas, split peas, lima beans, and pinto beans	20	140	↓ ERα protein/mRNA levels	Inhibits cell viability, cell growth, and proliferation ↑ Bax ↑ apoptosis ↑ cell cycle arrest ↑ ROS generation ↑ DNA damage ↑ ERK1/2 phosphorylation ↑ p53 proteins ↓ AKT phosphorylation	anti-cancer for breast and prostate cancers anti-tumor for ovarian, breast, lung, and cervical cancers decreased endometrial cancer risk	[98,99,100,101]

^a^ values adapted from Bowers et al. [75] were obtained using a different methodology; use with caution when making comparisons. ^b–d^ specific relative binding affinity values were not found.

**Table 3 ijms-22-08798-t003:** Clinical trials of phytoestrogens used as cancer prevention and/or cancer treatments.

Identifier	Cancer Type/Prevention	Chemicals	Date	Participants/ Type of Study	Aims	Results
NCT00597532 [226]	Breast	Genistein + Daidzein	8/2002–4/2016	140 women/ R P controlled study	To examine the effects of soy supplementation on breast cancer-related genes and pathways	Tumors- PRE vs POST = altered EXP of 21 out of 202 genes. ↑ FANCC & UGT2A1 EXP in TG vs. PG (*p* < 0.05) Over-EXP of FGFR2, E2F5, BUB1, CCNB2, MYBL2, CDK1, and CDC20 (*p* < 0.01) in tumors with high-genistein signature
NCT00513916 [[232],[233]	Breast	Isoflavones	7/2006–2/2012	82 multiethnic PR/ R, crossover ‡	To study the effects of dietary soy on estrogens in breast fluid, blood, and urine samples from healthy women	High-soy diet resulted in a modest trend of a lower cytological subclass in breast epithelial cells ↑isoprostane excretion in high-soy diet (*p* = 0.02)
NCT00612560 [229]	Breast	Ground flaxseed (FS) ± anastrozole (AI)	11/2007–4/2014	24 PO; 2 x 2 factorial R intervention	To examine the effect of flaxseed consumption compared to AI, and the effect of combined flaxseed and AI therapy on breast cancer treatment	↓ serum steroid hormone DHEA w/ AI treatment (*p* = 0.009) PRE vs POST in FS + AI = ~40% ↓ EXP of estrogen receptorβ Lower Enterolactone excretion in FS + AI vs FS
NCT00290758 [230]	Breast	Mixed soy isoflavones	1/2006–7/2009	126 women (≥ 25 years)/ R *B	To study how well genistein works in preventing breast cancer in women at high risk for breast cancer	↑ Ki-67 labeling index within PR TG (*p* = 0.04) Within TG, ↑ EXP of 14/28 genes (*p* = 0.017–0.052), but no S changes in PG TG vs PG = ↑ ESR1, FAS, FOXA1, MYB (*p* = NS)
NCT01219075 [231]	Breast	Daidzein, genistein, glycitein	7/2010–present	85 women (30–75 years)/ D-B, R, P-controlled	To study soy isoflavones supplement in treating women at high risk or with breast cancer	NS differences in breast density area (*p* = 0.23) and mammogrpahic density % (*p* = 0.38) in TG vs PG
University of North Dakota School of Medicine and Health Sciences [232]	Breast	*Trans*- resveratrol	N/A	39 women/D-B, R, P-controlled	To determine if trans-resveratrol had a dose-related effect on DNA methylation and prostaglandin expression in humans	↑ levels of trans-resveratrol & resveratrol-glucuronide in serum = ↓ RASSF-1α methylation (*p* = 0.047) & ↓ PGE2 EXP in breast (*p* = 0.045)
National Center of Oncology, Yerevan, Armenia (Ministry of Health Republic of Armenia Registration No.: Nr 5592-17-02-23) [233]	Breast	Curcumin + Paclitaxel	3/2017–9/2018	150 women (18–75 years)/ *, single-institution, R, P-controlled, D-B, parallel group, two-arm trial	To assess the efficacy and safety of intravenous curcumin infusion in combination with paclitaxel in patients with metastatic and advanced breast cancer	↑ objective response rate (ORR) of TG vs PG (16 weeks after beginning treatment, *p* < 0.05; completed treatment, *p* < 0.01) 3 months after treatment, ↑ ORR TG vs. PG (*p* < 0.07) ↑ fatigue in TG vs. PG (*p* = 0.05), but the overall physical performance was significantly higher with curcumin than the placebo during treatment and at the end of follow-up
NCT00596895 [234]	Prostate	soy milk containing isoflavonoid	11/2003–11/2007	20 men/ O-L, * nonrandomized trial	To evaluate the efficacy of isoflavone in patients with PSA recurrent prostate cancer after prior therapy.	Slope of PSA level (after vs. before study entry): ↓ in 6 men (*p* < 0.05), ↑ in 2 men (*p* < 0.05), and NS changes in 12 men A decline in the rise of serum PSA after the initiation of soy milk.
NCT01009736 [61]	Prostate	Tomato-soy juice	1/2008–7/2009	60 men/ * dose-escalating	To study the side effects of tomato-soy juice and its effect on biomarkers in patients with prostate cancer undergoing prostatectomy	High TG vs PG, ↓prostate-specific antigen slope (*p* = 0.078)
NCT00255125 [235]	Prostate	Soy isoflavone capsules	9/2005–10/2009	86 men (≥18 years)/ D-B, R, P-controlled	To evaluate the effects of soy isoflavone consumption on prostate specific antigen, hormone levels, total cholesterol, and apoptosis in men with localized prostate cancer.	TG vs PG in malignant prostate tissue = down-regulated 12 genes involved in cell cycle control and 9 genes involved in apoptosis No significant changes in serum total testosterone, free testosterone, total estrogen, estradiol, PSA, and total cholesterol
NCT00765479 [236]	Prostate	Soy protein isolate	9/2011–7/2013	284 men (40–75 years)/ R, P-controlled	Secondary analysis of body weight, blood pressure, thyroid hormones, iron status, and clinical chemistry in a 2-y trial of soy protein supplementation in middle-aged to older men.	Soy supplementation did not affect body weight, blood pressure, serum total cholesterol, iron status parameters, calcium, phosphorus, and thyroid hormones.
NCT00546039 [237]	Prostate	Synthetic genistein	4/2007–1/2009	47 Norwegian men/ * P-controlled, R, D-B	To evaluate safety and mechanisms of possible chemopreventive effects of synthetic genistein (BONISTEIN) in patients with localized prostate cancer undergoing laparoscopic radical prostatectomy	Genistein intervention significantly reduced the mRNA level of KLK4 in tumor cells (*p* = 0.033) and p27Kip1 In genistein intervention, no significant effects on proliferation-, cell cycle-, apoptosis-, or neuroendocrine biomarkers
NCT02724618 [238]	Prostate	Nanocurcumin	3/2016–present	64 men/ R, D-B, * P-controlled	To determine the efficacy of oral nanocurcumin in prostate cancer patients undergoing radiotherapy.	Nanocurcumin was well tolerated. No significant difference was found between two groups regarding tumor response.
NCT02138955 [239]	Prostate, Colon	Curcumin	3/2014–6/2017	32 participants (18–85 years)/ ∞, single-center, O-L	To investigate the safety and tolerability of increasing doses of liposomal curcumin in patients with metastatic cancer	300 mg/m^2^ liposomal curcumin over 6 h was the maximum tolerated dose, and a recommended starting dose for anti-cancer trials Anti-tumor activity was not detected
NCT01917890 [240]	Prostate	Curcumin	3/2011–10/2013	40 men (50–80 years)/ R, D-B, P-controlled	To evaluate the effect of curcumin supplementation on oxidative status of patients with prostate cancer who undergo radiotherapy	In TG: ↓ activity of superoxide dismutase (SOD) (*p* = 0.026), and ↑ plasma total antioxidant capacity (TAC) (*p* = 0.014) ↓ PSA level in both TG and PG No significant differences in treatment outcomes were observed between TG and PG
NCT00118846 [241]	Endometria	Genistein, daidzein, glycitein,	3/2004–3/2009	350 women (45-92 years)/ R, D-B, P-controlled	To determine whether long-term isoflavone soy protein (ISP) supplementation affects endometrial thickness and rates of endometrial hyperplasia and cancer in postmenopausal women	Soy-treated group did not significantly differ on the mean baseline or on-trial changes in endometrial thickness ISP has been found to predominantly act on the beta-type estrogen receptor because of its structure similar to 17β-estradiol and selective estrogen receptor modulator (SERM)-like activity.
NCT02017353 [242]	Endometrial	Curcumin Phytosome (CP)	10/2013–10/2016	7 women (≥18 years)/ O-L, * non-randomized	To determine whether curcumin can inhibit tumor induced inflammation in patients with endometrial carcinoma. In addition, curcumin could possibly induce a better functioning of chemotherapy and a decrease in toxicity from chemotherapy.	In TG, downregulated MHC expression levels on leukocytes (*p* = 0.0313), frequency of monocytes (*p*= 0.0114), and ICOS expression by CD8+ T cells (*p* = 0.0002), but upregulated CD69 levels on CD16- NK cells (0.0313).
NCT00256334 [243]	Colon	Trans-resveratrol + quercetin	7/2005–4/2009	11 participants (≥18 years)/∞ pilot, O-L	To evaluate the effects of a low dose of plant-derived resveratrol formulation and resveratrol-containing freeze-dried grape powder on Wnt signaling in the colon	Resveratrol did not inhibit Wnt pathway in colon cancer, but did inhibit Wnt pathway in normal colonic mucosa (*p* < 0.03)

R, randomized; D-B, double-blind; P, placebo; O-L, open-label; ∞, phase I; *, phase II; ‡, phase III; TG, treatment group; PG, placebo group; PRE, pre-treatment; POST, post-treatment; PR, premenopausal women; PO, postmenopausal women; B, baseline; NS, non-significant; S, significant; EXP, expression.

## Data Availability

Not applicable.

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
