# Peer review of "Exploring the Biological Activity and Mechanism of Xenoestrogens and Phytoestrogens in Cancers: Emerging Methods and Concepts"

_ijms, 2021, doi:10.3390/ijms22168798_

Round 1

Reviewer 1 Report

The manuscript by Xiaoqiang Wang et al. describes the biological activity and mechanism of Xenoestrogens and Phytoestrogens in human physiology. The authors presented an interesting review, well written with a clear aim. Noteworthy, the topic is very important, because the environmental aspects of cancer pathogenesis are still very uncertain. Moreover, estrogen-dependent cancers: breast cancer, and colorectal cancer are very common. 

Strong points of the article:

  1. Important and “trendy” topic
  2. Clearly written
  3. Very good introduction
  4. The claims are convincing and supported by the literature

However, there are some issues that could be explained/discussed, or corrected:

  1. Abstract:

(A) In my opinion, the abstract emphasize just selected points without keeping a balance. Additionally, it does not give a full picture of the article. Not mentioning 1/3 of the article (chapter 4), which is biological activity (A) Xenoestrogens and Phytoestrogens.

(B) what the emerging concept of WOS?  I'm wondering if the “WOS” is not just a new nomenclature fact previously called critical developmental windows or sensitive window?

To sum up, an abstract should be a summary of the manuscript however, the current one requires improvement.

  1. Main text

(A) In general description of Xenoestrogens, we have together commonly used bisphenol A and currently “historical” DDT, and I believe that either the text content or the order this could more objectively separate/compare the importance of these points.

(B) Authors are focusing on cancer however, other health problems should be at least mentioned (infertility, reproductive hormones, or thyroid function).

(C) I'm wondering if some other articles describing patient‐derived xenografts (PDX) and endocrine disruptors might be discussed i.e:

(A) Fahmi Mesmar, 1 , 2 , 9 Bingbing Dai, 3 Ahmed Ibrahim, 2 Linnea Hases, 2 , 4 Mohammed Hakim Jafferali, 2 Jithesh Jose Augustine, 3 Sebastian DiLorenzo, 5 , 6 Ya'an Kang, 3 Yang Zhao, 7 Jing Wang, 7 Michael Kim, 3 Chin‐Yo Lin, 1 Anders Berkenstam, 8 , 10 Jason Fleming, 3 , 11 and Cecilia Williams Clinical candidate and genistein analogue AXP107‐11 has chemoenhancing functions in pancreatic adenocarcinoma through G protein‐coupled estrogen receptor signaling. Cancer Med. 2019 Dec; 8(18): 7705–7719.

(B) M Angeles Lillo 1 2, Cydney Nichols 3, Chanel Perry 1 2, Stephanie Runke 4, Raisa Krutilina 5 2, Tiffany N Seagroves 5 2, Gustavo A Miranda-Carboni 6 2, Susan A Krum. Methylparaben stimulates tumor initiating cells in ER+ breast cancer models. J Appl Toxicol. 2017 Apr;37(4):417-425. doi: 10.1002/jat.3374. Epub 2016 Sep 1.

Author Response

  1. Abstract:

(A) In my opinion, the abstract emphasizes just selected points without keeping a balance. Additionally, it does not give a full picture of the article. Not mentioning 1/3 of the article (chapter 4), which is biological activity (A) Xenoestrogens and Phytoestrogens.

Authors: Thank you for the constructive recommendations on our Abstract. We thus rewrote the abstract as suggested. The revised abstract is on Page 2, lines 8-20

(B) what the emerging concept of WOS?  I'm wondering if the “WOS” is not just a new nomenclature fact previously called critical developmental windows or sensitive window?

Authors: The Endocrine Society (https://www.hormone.org/your-health-and-hormones/endocrine-disrupting-chemicals-edcs) defines “sensitive windows of development” as the periods of fetal development and childhood where hormones regulate the formation and maturation of organs. Therefore, early-life exposures have been linked to developmental abnormalities and may increase the risk for a variety of diseases later-in-life.

According to the definition by The Breast Cancer and the Environment Program (BCERP) (https://bcerp.org/researchers/windows-of-susceptibility/), “windows of susceptibility/WOS” covers a wide lifespan spectrum, including the prenatal, pubertal, pregnancy, and menopausal transition periods, during which the mammary glands undergo anatomical and functional transformations. Therefore, environment hormones (e.g., endocrine-disrupting chemicals (EDC)) and certain therapeutics (e.g., prescribed for the coexisting medical conditions or in the form of the hormone replacement therapy) can act upon breast tissue during particular WOS and influence breast cancer risk, development, or outcome.

Therefore, these two concepts are different because they emphasize different, distinct life stages in the human lifecycle. We clarified the concept of WOS in the abstract and in the main text as well. These changes have been made on Page 2, lines 14-17, Page 4, lines 15-24, Page 17, lines 1-3 and Page 22, lines 2-12.

To sum up, an abstract should be a summary of the manuscript however, the current one requires improvement.

Authors: Thank you for the constructive recommendations. We thus improved the abstract as suggested above. The revised abstract is on Page 2, lines 8-20.

  1. Main text

(A) In general description of Xenoestrogens, we have together commonly used bisphenol A and currently “historical” DDT, and I believe that either the text content or the order this could more objectively separate/compare the importance of these points.

Authors: Thank you for your advice. We have adjusted the order of BPA and DDT and addressed their current public health concerns. These changes were made on Page 5, lines 13-46.

(B) Authors are focusing on cancer however, other health problems should be at least mentioned (infertility, reproductive hormones, or thyroid function).

Authors: Thank you for your suggestions. We extended the introductions on the etiology of reproductive health and non-reproductive disorders. These changes were made on Page 17, lines 28-43, Page 18, lines 1-12.

(C) I'm wondering if some other articles describing patient‐derived xenografts (PDX) and endocrine disruptors might be discussed i.e:

Authors: Thank you for your suggestion. We included the first reference from your suggestions and explained their results [128]. The second suggested reference investigating the effect of methylparaben was already been included in [123]. We also found a paper from Kimura et al. which showed a preliminary result in a prostate PDX model treated with diethylstilbestrol (DES). This paper was included in [127]. Although there were some papers using PDX models with the compounds explained in our manuscripts, most of them did not focus on their estrogenic activity. Therefore, we did not include them in our manuscript. We made these changes on Page 10, line 26-29.

Reviewer 2 Report

In the manuscript, "Exploring the Biological Activity and Mechanism of Xenoestrogens and Phytoestrogens in Cancers: Emerging Methods and Concepts', authors made a good attempt on the role of external estrogens in the development of cancer. Even though the concept of the study is good, the execution was very poor. 

The major concern of this study is that most of the Xenoestrogens are known to induce tumorigenesis. Most of the Phytoestrogens are known to inhibit cancer development, and many of those Phytoestrogens are very common anti-cancer candidate drugs, and their molecular mechanism is also well known. Hence a common conclusion for these different estrogen does not make sense. 

The authors failed to convey the exact aim of this study as most of the conclusions were became out of focus or non-relevant to the given conclusion. 

Altogether, this manuscript does not suitable for publication in this current format. 

Author Response

Authors: Thank you for your dialectic and constructive comments. To better acknowledge the potential benefits of phytoestrogens in the prevention and/or treatment of cancers, we reviewed 18 clinical trials, conducted between 2002 to present, that were focused on breast cancer (7 trials), prostate cancer (8 trials), colon cancer (2 trials), and endometrial cancer (2 trials), combined with 2 categories of phytoestrogens treatments: fruits/whole grains/seeds such as resveratrol and curcumin (8 trials), and soy isoflavones such as genistein and genistein (10 trials). We summarized the data from these clinical trials in Section 5 (Pages 19-20), Table 3 & Supplementary Table 1.

Existing data generally support the safety of small doses of purified phytoestrogen consumption as a medication regarding the breast cancer risk. However, for the general population, including women with benign breast disorders, those at risk for breast cancer, and even survivors of cancer, the prescription of phytoestrogens is still not recommended due to insufficient evidence.  Although not all 18 trials exhibited promising results, patients receiving curcumin combined with a chemotherapy drug, Paclitaxel, to treat metastatic breast cancers had a greater objective response rate (p <0.05 16 weeks after starting treatment, and p < 0.01 after completed treatment) compared to patients receiving Paclitaxel alone. Moreover, some men with prostate cancer had a slow rise of serum PSA after consuming 141 mg of isoflavones per day. This prostate cancer trial has also shown that genistein may have an inhibitory effect on androgen-related biomarkers and supports genistein as a chemo-preventive agent in prostate cancer.

More recent clinical trials have used gene expression analysis to study phytoestrogens and their response to breast cancer, prostate cancer, endometrial cancer, and colon cancer. More interestingly, these trials have shown that phytoestrogens alter cancer-related gene expression profiles. Under the concept of personalized medicine, gene expression analysis could be an alternative and cost-effective way to predict the effectiveness of phytoestrogens in cancer prevention and treatments. However, a larger number of clinical trial participants and more studies of phytoestrogens and their impact on cancers are still needed to better define their potential.

We have added a new section titled “Application of Phytoestrogens in the Prevention or Treatment of Cancers: Evidence from Clinical Trials” that also includes two tables (Table 3 & Supplementary Table 1). We also carefully rewrote the context of phytoestrogens in “Future Directions and Conclusions”, by summarizing what we learn from the clinical trials and providing a perspective for future studies. We made these changes on Page 19 & 20, and Page 23, lines 1-13.  

Round 2

Reviewer 2 Report

The authors have made a good attempt to improve the quality of the manuscript. Considering the addition of a new section, the structure of this review is quite good. 

I do not have any further concerns.